# ZIP-FIT: Embedding-Free Data Selection via Compression-Based Alignment for Code

## Abstract

Data selection is crucial for optimizing language model (LM) performance on specific tasks, yet most existing methods fail to effectively consider the target task distribution. Current approaches either ignore task-specific requirements entirely or rely on approximations that fail to capture the nuanced patterns needed for tasks like Autoformalization or code generation. We introduce `ZIP-FIT`, a data selection framework that uses compression to directly measure the alignment between potential training data and the target task distribution. Our key insight is that compression-based similarity captures both syntactic and structural patterns relevant to the target task (like code), enabling more precise selection of task-relevant data for code. In extensive evaluations on Autoformalization and Python code generation, `ZIP-FIT` significantly outperforms leading baselines like DSIR and D4. Models trained on `ZIP-FIT`-selected data achieve their lowest cross-entropy loss up to 85.1% faster than these baselines, demonstrating that better task alignment leads to more efficient learning. In addition, `ZIP-FIT` performs selection up to 65.8% faster than DSIR and two orders of magnitude faster than D4. In addition, we achieve 18.86% Pass@1 on HumanEval compared to LESS's 18.06% while being approximately 2000 times faster. Notably, `ZIP-FIT` shows that smaller, well-aligned datasets often outperform larger but less targeted ones, demonstrating that a small amount of higher quality data is superior to a large amount of lower quality data.

## 1 Introduction

Choosing training data is crucial for the performance of language models (LMs) in both general-purpose and domain-specific applications (Brown et al., 2020; Gururangan et al., 2020; Hoffmann et al., 2022). To date, most research on data curation has focused on creating diverse pre-training datasets to enhance model performance across a wide range of tasks (Sorscher et al., 2022; Xie et al., 2023b; Tirumala et al., 2023; Abbas et al., 2023; Xie et al., 2023a; Lee et al., 2023; Wettig et al., 2024; Penedo et al., 2024; Li et al., 2024; Sachdeva et al., 2024), and while these methods have been demonstrated to work well for general pre-training, they fall short in domain-specific fine-tuning, where data relevance is crucial. This raises a key question: *Is there an effective methodology we can use to effectively select fine-tuning data for domain-specific target tasks such as coding?*

One approach is to train classifiers to identify relevant data. For example, DeepSeekMath (Shao et al., 2024) used a compilation of high-quality mathematical texts (Paster et al., 2023), to train a classifier to retrieve similar texts from the Web (Bojanowski et al., 2017). This method depends on large, well-annotated datasets, which are often unavailable for niche tasks.

Embedding-based methods measure similarity between data points and a reference corpus (Xie et al., 2023c), selecting relevant data, but incurring high computational costs and being highly dependent on the choice of embedding space (Muennighoff, 2022). DSIR (Data Selection via Importance Resampling) (Xie et al., 2023b) instead utilizes unigrams and bigrams to select data points without the need for pre-trained embeddings, with the aim of matching the hashed n-gram distributions of the target data. Although DSIR is effective in capturing direct word correlations, it may not capture structured patterns of syntax that unfold across sentences or paragraphs, such as nested function calls in code or embedded clauses in formal language translation (Moura et al., 2015). Additionally, the hashing introduces noise due to collisions.

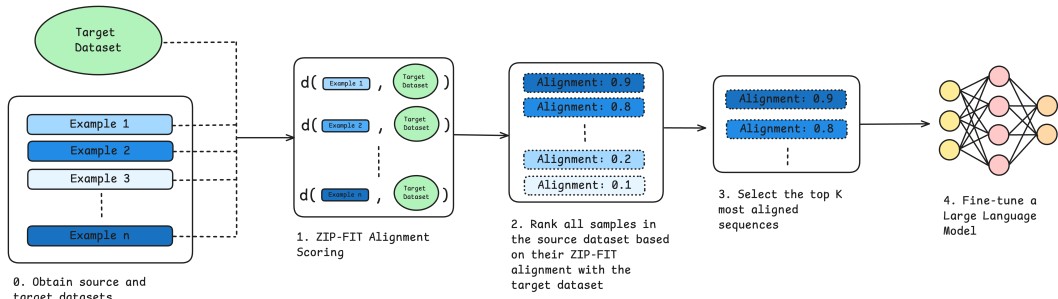

Figure 1: **`ZIP-FIT` selects task-specific data for efficient finetuning.** (0) Obtain both the source and target datasets. (1) Calculate `ZIP-FIT` Alignment of each source example with the target dataset using compression. (2) Rank all source examples based on these alignment scores. (3) Select the top-K most aligned examples for fine-tuning. (4) Fine-tune a large language model using the selected top-K examples to improve performance on the target task.

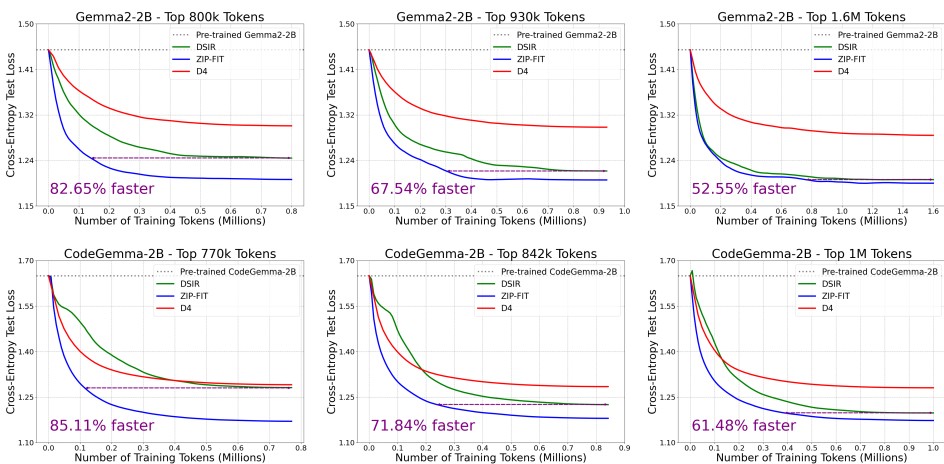

Figure 2: **Code Generation: `ZIP-FIT` accelerates cross-entropy loss reduction, even in code-specialized models like CodeGemma-2B.** The plots show cross-entropy test loss versus the number of training tokens for Gemma-2-2B (top row) and CodeGemma-2B (bottom row) across different token selection sizes. `ZIP-FIT` (blue) consistently reduces loss faster than DSIR (green) and D4 (red), achieving up to $85.11\%$ speed improvement at lower token counts. These results demonstrate `ZIP-FIT`'s efficiency in data selection for fine-tuning models on code-geneation tasks.

Gradient-based methods like LESS (**L**ow-rank gradi**E**nt **S**imilarity **S**earch) (Xia et al., 2024) take a different approach, selecting data with gradients most similar to the target data. While LESS can identify useful training examples, computing and storing gradient features adds computational overhead. These limitations highlight the need for better domain-specific data selection strategies.

To address these challenges, we propose `ZIP-FIT`, a novel data selection framework that leverages compression algorithms (e.g., gzip, lz4, etc.). Recent research suggests that language modeling and data compression are fundamentally equivalent tasks (Delétang et al., 2024), and the intelligence of large language models (LLMs) is closely related to their ability to compress external corpora (Huang et al., 2024). This insight suggests that compression algorithms can encode information in ways similar to neural networks. For example, Jiang et al. (2023c) found that the use of normalized compression distances for text classification outperformed traditional neural embeddings. Inspired by this, `ZIP-FIT` selects aligned training data with a target data set based on compression-based alignment, providing a lightweight and embedding-free method for selecting high-quality data.

We evaluated `ZIP-FIT` in two domains: Autoformalization and Python code generation. `ZIP-FIT` outperforms existing data selection methods, consistently improving model performance across

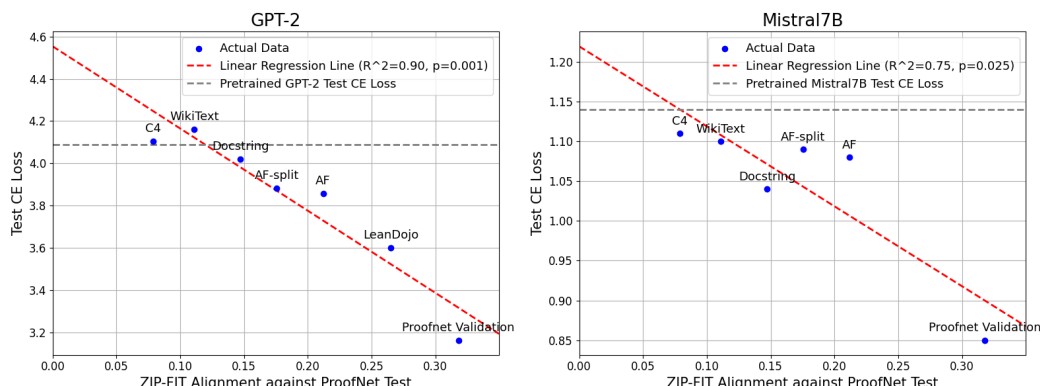

Figure 3: **Higher `ZIP-FIT` alignment correlates with lower cross-entropy loss.** The relationship between `ZIP-FIT` alignment and cross-entropy (CE) loss for (a) GPT-2 trained on 22k tokens ($R^2 = 0.90, p = 0.001$) and (b) Mistral7B trained on 22k tokens ($R^2 = 0.75, p = 0.025$). Each point represents a dataset, with its position reflecting the dataset's `ZIP-FIT` alignment score against the ProofNet validation split and the resulting testnCE loss. The dashed red line indicates the linear regression fit, while the dashed grey line shows the pretrained CE loss. Higher alignment scores correspond to lower CE losses, demonstrating that training on better aligned data yields better performance.

multiple metrics. Smaller, well-aligned datasets selected by `ZIP-FIT` lead to faster convergence and better performance than larger, less aligned datasets, highlighting the importance of data quality.

Our **contributions** are as follows:

1. **Methodology:** The introduction of `ZIP-FIT`, an embedding-free data selection method based on compression.

2. **Superior Performance:** `ZIP-FIT` achieves faster convergence (up to 85.1%) and lower test cross-entropy loss compared to data selection baselines DSIR and D4. On downstream task metrics, `ZIP-FIT` outperforms DSIR and LESS in autoformalization, measured with syntax error compilation Pass@k, and exceeds DSIR, D4, and LESS on HumanEval, evaluated with the unit test-based Pass@k metric.

3. **Computational Efficiency:** `ZIP-FIT` is computationally efficient, running up to 65.8% faster than DSIR. This makes it scalable for low-resource environments without compromising performance.

## 2 `ZIP-FIT`: AN EMBEDDING-FREE DATA SELECTION ALGORITHM VIA COMPRESSION-BASED ALIGNMENT FOR LM FINE-TUNING

Before introducing `ZIP-FIT`, it is essential to understand the desired attributes of effective data selection algorithms. Ideally, such algorithms should be performant, computationally economical, fast, scalable, and designed to improve the efficiency of model training. These characteristics ensure that the data filtering process can be applied broadly and effectively in various machine learning contexts, particularly when computational resources are limited. By setting these criteria, we can better appreciate the innovations `ZIP-FIT` introduces in the realm of data selection.

### 2.1 BACKGROUND

**Lossless Compression**: Lossless text compression algorithms reduce data size by exploiting statistical redundancies while ensuring perfect reconstruction of the original data. Pattern-based techniques like LZ77 identify and replace repeated subsequences with compact references, while statistical encoding methods such as Huffman coding assign shorter codes to frequent symbols. For more details, see Appendix B.

**Autoformalization:** The task of translating natural language mathematics into formal programming languages like Lean4 Moura et al. (2015). This process requires precise understanding and representation of mathematical formal syntax, making the selection of well-aligned training data crucial for effective model training.

## 2.2 ZIP-FIT ALGORITHM

**Setup:** Given a set of examples $\{x'_1, x'_2, \ldots, x'_n\}$ from a target distribution $p$ and a large source dataset $\{x_1, x_2, \ldots, x_N\}$ from an arbitrary distribution $q$, ZIP-FIT aims to select a subset of K examples (where $K \ll N$) from $q$. The selected subset is used for model training, in order to improve performance for tasks associated with $p$. This approach is intended to maximize the efficacy and efficiency of model training by focusing on the most relevant data samples.

**Method:** ZIP-FIT uses compression as a metric to measure the alignment of each example in $q$ with the target $p$, focusing on capturing patterns and redundancies.

To address the challenge of selecting highly aligned data, we propose the ZIP-FIT algorithm:

[H] [1] **Input:** A source dataset $D = \{x_1, x_2, \ldots, x_N\}$ from distribution $q$, target examples $\{x'_1, x'_2, \ldots, x'_n\}$ from distribution $p$. **Output:** A subset of K examples from $D$ that improve performance for $p$. $i = 1$ **to** $N$ Compute alignment for each sample $x_i \in D$ with each target example $x'_j \in \{x'_1, x'_2, \ldots, x'_n\}$ using Normalized Compression Distance:

$$\text{NCD}(x_i, x'_j) = \frac{C(x_i \oplus x'_j) - \min(C(x_i), C(x'_j))}{\max(C(x_i), C(x'_j))}$$

where $C(x)$ represents the compressed size of sequence $x$ and $\oplus$ denotes concatenation. Compute the average ZIP-FIT alignment for each $x_i$:

$$\text{ZIP-FIT-Alignment}(x_i) = 1 - \frac{1}{n} \sum_{j=1}^{n} \text{NCD}(x_i, x'_j)$$

Select the top-K examples from $D$ based on the highest alignment scores.

## 3 HIGHER ZIP-FIT ALIGNMENT CORRELATES WITH BETTER MODEL PERFORMANCE

**Experiment:** To validate the effectiveness of compression-based alignment, we evaluate the relationship between ZIP-FIT alignment scores and model performance on the target task. Using ProofNet's validation split as the target distribution and test split for evaluation, we fine-tune GPT-2 Radford et al. (2019) and Mistral7B Jiang et al. (2023b) on datasets with varying ZIP-FIT alignment scores. To ensure fair comparison, we standardize each dataset to 100k tokens, except where datasets are inherently smaller (e.g., ProofNet validation set).

**Results** Figure 3 demonstrates a strong negative correlation between ZIP-FIT alignment scores and cross-entropy (CE) loss, with $R^2$ values of 0.90 and 0.75 for GPT-2 and Mistral7B respectively. Highly-aligned datasets like LeanDojo Yang et al. (2023) and the ProofNet validation split yield substantially lower CE loss compared to less-aligned datasets like C4 Raffel et al. (2020) and WikiText Merity et al. (2016). This suggests that ZIP-FIT effectively identifies training data that improves model performance on the target task.

## 4 COMPARATIVE EVALUATION OF ZIP-FIT FOR EFFICIENT FINE-TUNING

We evaluate ZIP-FIT on two domain-specific tasks: *Autoformalization* and *Python Code Generation*. Our goal is to show ZIP-FIT's data selection leads to superior fine-tuning performance for coding tasks compared to baselines.

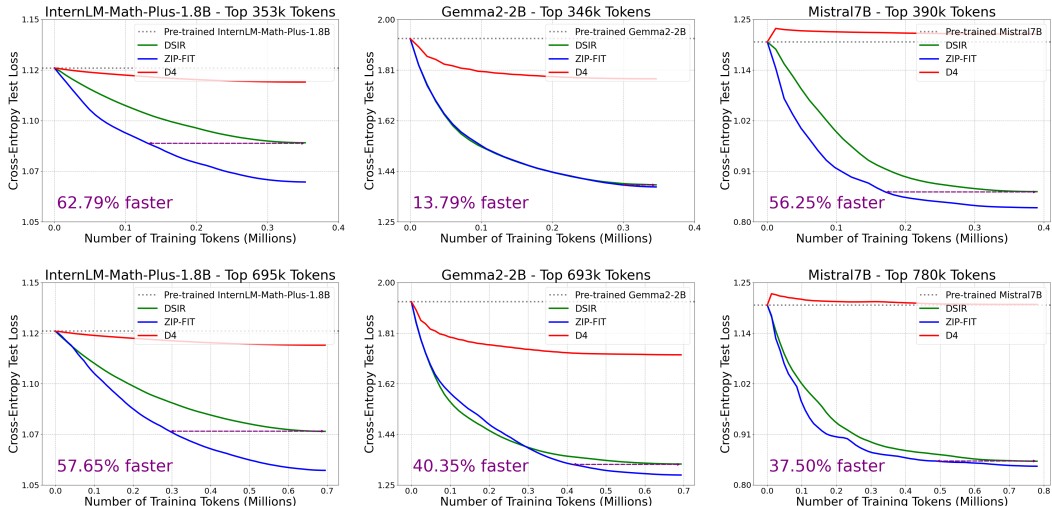

Figure 4: **AutoFormalization: `ZIP-FIT` consistently achieves lower test loss more quickly than D4 and `DSIR`, demonstrating its efficiency in data selection.** The plots show cross-entropy test loss versus the number of training tokens for three models (InterLM-Math-Plus-1.8B, Gemma-2-2B, and Mistral7B) across different token selection sizes. `ZIP-FIT` (blue line) consistently outperforms both `DSIR` (green line) and D4 (red line) across all model and token size configurations, highlighting its ability to process data more efficiently. The percentage labels in each plot indicate the relative speedup of `ZIP-FIT` over `DSIR` in reaching the lowest cross-entropy loss, reinforcing the method's scalability and adaptability for domain-specific fine-tuning.

## 4.1 CODE GENERATION

**Experiment:** We evaluated data selection methods for Python code generation using `ZIP-FIT`, DSIR, and D4 on datasets combining MBPP Austin et al. (2021), Python docstrings, Proof-Pile 2, C4, and WikiText (composition details: Appendix B.3). Alignment scores were computed using HumanEval's validation split, with top-$n$ sequences selected (800K, 930K, 1.6M tokens) without modifying baseline ranking mechanisms. Cross-entropy (CE) loss was evaluated on CodeGemma-2B and Gemma-2-2B, while HumanEval Pass@1 results focus on Gemma-2-2B with added comparisons to LESS. We use a separate hold-out portion for final testing.

**Results:** `ZIP-FIT` accelerates convergence and improves code generation quality. For CE loss, `ZIP-FIT` achieves 85.1% faster convergence than DSIR on CodeGemma-2B and 67.5% faster convergence on Gemma-2-2B. D4, again, demonstrates poor performance.

On HumanEval Pass@1 (Table **??**), `ZIP-FIT` achieves 18.86% accuracy, outperforming DSIR (17.98%) and LESS (18.06%) while requiring 3× less time than DSIR and no GPUs. LESS's computational cost (19h on 4 GPUs) limits its practicality.

## 4.2 AUTOFORMALIZATION

**Experiment:** Our source dataset comprised approximately 185,000 sequences from LeanDojo, Proof-Pile 2, C4, and WikiText Yang et al. (2023); Azerbayev et al. (2024). For details on dataset composition, refer to Appendix B.2. Training datasets were curated using `ZIP-FIT`, DSIR, and D4, with ProofNet's validation split as the target distribution. For fairness, we preserved the native ranking mechanism of each method and selected top-n sequences (e.g., 353K, 695K tokens). We fine-tuned three models—InternLM-Math-Plus-1.8B Ying et al. (2024), Gemma-2-2B Team et al. (2024), and Mistral-7B—on each subset, evaluating cross-entropy (CE) loss on ProofNet's test split.

**Results:** Figure 4 shows that `ZIP-FIT` significantly outperforms DSIR and D4 in reducing cross-entropy (CE) loss across all token selection sizes (353k, 695k). The steep decline in the blue curves (`ZIP-FIT`) highlights its ability to achieve faster convergence, resulting in up to 62.79%

| Method | Pass@1 (%) | Time | Hardware |
|--------|-----------|------|----------|
| Gemma-2-2B | 15.24 | – | – |
| ZIP-FIT | **18.86** | **32s** | CPU |
| DSIR | 17.98 | 97s | CPU |
| LESS | 18.06 | 19h | 4 A100-80GB |
| D4 | 14.37 | 7h 40m | 1 A100-80GB |

Table 1: **Python Code Generation**: Comparison of data selection methods for code generation using Gemma-2-2B. Pass@1 accuracy on HumanEval, selection time, and hardware requirements are shown. ZIP-FIT achieves the highest Pass@1 score with the fastest selection time using only CPU resources. All models where fine-tunes of Gemma-2-2B. For all methods, the top 1M tokens were selected

improvements in convergence speeds compared to DSIR. Notably, `ZIP-FIT` demonstrates up to a 65.8% faster data selection process than DSIR. Similar results were observed at other token counts, as detailed in Appendix E.

Additionally, we use a syntax error compilation Pass@k metric. Given $k$ samples, we consider a trial successful if at least one sample compiles without syntax errors. This method ensures that generated formal statements are at least syntactically valid in Lean4, even if they are not fully correct. (Aniva et al., 2024) Table **??** shows the results for Pass@5 across 183 tasks. The exact prompt format used is detailed in Appendix K.

| Method | Pass@5 (%) | Time | Hardware |
|--------|-----------|------|----------|
| Gemma-2-2B | 6.5 | – | – |
| ZIP-FIT | **14.0** | **79s** | CPU |
| DSIR | 0.0 | 135s | CPU |
| LESS | 0.1 | 20h 45m | 4 A100-80GB |

Table 2: **AutoFormalization**: `ZIP-FIT` achieves higher Lean4 compilation pass @ 5 than competing methods on ProofNet's test split. We computed the number of Lean4 compilation passes with k=5 samples. All models where fine-tunes of Gemma-2-2B. For all methods, the top 1M tokens were selected

Notably, `ZIP-FIT` achieves a Pass@5 score of 14.0%, more than double that of the base Gemma-2-2b model (6.5%) and orders of magnitude higher than prior data selection methods—DSIR (0.0%) and LESS (0.1%). These results indicate that ZIP-FIT is particularly effective at enhancing the model's ability to generate syntactically valid and semantically meaningful Lean4 code for autoformalization tasks. The fact that DSIR and LESS yield virtually no valid completions suggests that they may be selecting data poorly suited to the rigid formal syntax of Lean, thereby failing to support model learning in this highly structured domain. Crucially, ZIP-FIT achieves this performance without any GPU acceleration, running entirely on CPUs—whereas LESS, despite requiring 4× A100-80GB GPUs, fails to produce usable improvements. This highlights not only the effectiveness but also the efficiency and practicality of ZIP-FIT as a fine-tuning data selection method for code.

## 5 IMPACT OF DATA MISALIGNMENT ON MODEL PERFORMANCE

Existing research showed that data alignment plays a critical role in improving model performance and learning efficiency for downstream tasks. In this section, we explore how misalignment in data can affect model performance and how `ZIP-FIT` addresses this issue with data selection.

**Experiment:** We fine-tuned the Mistral7B model on the same source dataset we used for the Auto-Formalization experiment (see Appendix 4.2), filtering data with `ZIP-FIT` at different alignment thresholds ($>0.1$, $>0.2$, $>0.3$). Each threshold creates a progressively more aligned dataset, where the $>0.3$ dataset is the most aligned, and the $>0.2$ dataset is a superset of the $>0.3$ dataset, including

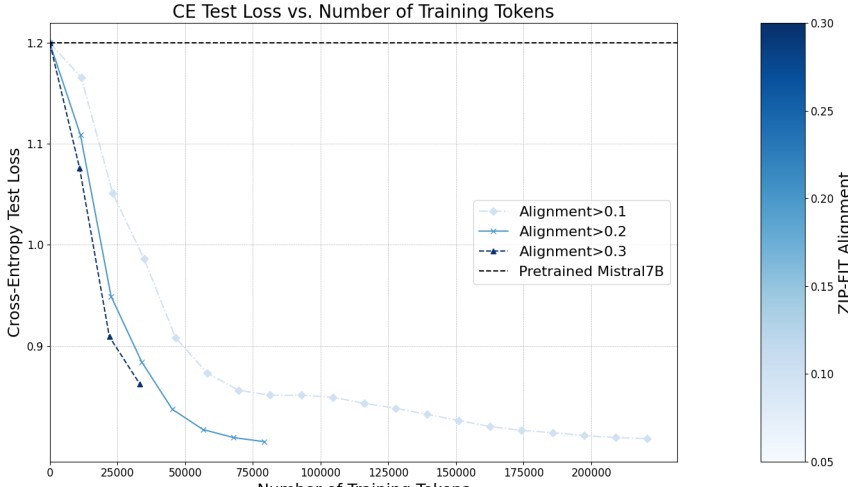

Figure 5: **Selective data filtering with `ZIP-FIT` allows us to achieve better cross-entropy test loss faster than training on all the data, resulting in improved performance and efficiency.** The x-axis represents the number of training tokens, while the y-axis shows the cross-entropy test loss. The curves represent models fine-tuned (FT) on datasets filtered by varying alignment thresholds ($>$ 0.1, $>$ 0.2, $>$ 0.3). The dashed line indicates the baseline performance of the pretrained Mistral7B model. Training on data filtered with higher alignment thresholds leads to superior performance, demonstrating the effectiveness of removing misaligned data in fine-tuning.

less aligned data. Similarly, the $>$0.1 dataset is a superset of both $>$0.2 and $>$0.3. Figure 5 shows CE test loss (y-axis) versus the number of training tokens (x-axis).

**Results:** `ZIP-FIT` selected data achieves lower CE loss faster than training on all data (Figure 5), showing improved performance and efficiency. Higher alignment thresholds result in a steeper loss reduction, confirming that *filtering out misaligned data enhances fine-tuning*. Misalignment can introduce noise and irrelevant patterns, which we hypothesize require more training data and computational resources to overcome. Applying higher alignment thresholds, `ZIP-FIT` ensures that only the most relevant examples are used for training. This targeted selection leads to a more efficient learning process as evidenced by the sharper decline in CE loss for higher alignment thresholds. Such efficiency is crucial in scenarios where computational resources are limited or costly.

**Practical Considerations:** For practitioners, these results suggest that investing in better data curation and alignment tools can significantly cut down the cost and time of model training without compromising performance. It also highlights the potential pitfalls of using large, uncurated datasets that might slow down the learning process or lead to poorer generalization on specific tasks.

**Future Directions:** Could explore adaptive alignment thresholds based on real-time validation CE, potentially automating the selection process to optimize both speed and accuracy during training.

Filtering out misaligned data accelerates fine-tuning and reduces computational overhead, confirming its performance gains and computational efficiency as outlined in our contributions.

## 6 RELATED WORKS

**Curating pre-training data** often involves using classifiers to filter high-quality data from large corpora like Common Crawl, as done for models like GPT-3 and PaLM2 (Brown et al., 2020; Google, 2023; Shao et al., 2024). While effective, this process requires significant computational resources and large volumes of curated data. In contrast, `ZIP-FIT` efficiently selects relevant data without relying on external models, making it especially useful in data-scarce environments.

**Deduplication** techniques, such as SemDeDup (Abbas et al., 2023) and D4 (Tirumala et al., 2023), improve data efficiency by removing duplicate or semantically similar examples using embedding-

based clustering. However, these methods are computationally expensive and not tuned to the target task. `ZIP-FIT` is embedding-free and task-aware, making it both scalable and more effective at selecting relevant data.

**Mixture weights** are essential when drawing from multiple domains, as they influence the performance of language models (Du et al., 2022; Xie et al., 2023b). DoReMi (Domain Reweighting with Minimax Optimization) (Xie et al., 2023a) proposes a reweighting strategy suitable for handling diverse target distributions, but it primarily focuses on adjusting weights at the domain level. Adapting it to select individual data points for specific target distributions would require substantial modifications to its foundational algorithm. One potential approach would be to effectively transform each data point into a 'mini-domain,' a process that would stray significantly from DoReMi's original purpose and scope. Therefore, we did not use DoReMi in our comparisons because it does not directly address the fine-grained selection needs that `ZIP-FIT` fulfills.

**Autoformalization** refers to the process of translating natural language mathematics into formal language (Wang et al., 2020; Wu et al., 2022), which is advantageous because formal proofs can be verified for correctness. However, the ability of current models to autoformalize text is limited by the scarcity of human-curated formal data. `ZIP-FIT` provides a framework for selecting the most relevant data, ensuring that models are trained on aligned datasets that enhance their performance. ]

**Compression:** (Pandey, 2024) demonstrates that gzip-compressibility predicts how language model scaling laws shift with data complexity, challenging the data-agnostic assumptions of Chinchilla-style scaling laws. (Jiang et al., 2022) use `gzip`-based compression distance with kNN for zero-shot text classification, outperforming BERT on some datasets. (Delétang et al., 2024) show that language modeling is equivalent to compression, where even `gzip` can define predictive distributions via coding length. (Yoran et al., 2025) propose the KoLMogorov Test, showing that `gzip` remains a strong baseline for code-based compression, outperforming pretrained LLMs on real data. show that `gzip` compression ratio is a fast, effective proxy for text diversity, capturing key repetition patterns in LLM outputs and aligning with slower lexical metrics e.g., self-BLEU.

# 7 LIMITATIONS

While `ZIP-FIT` provides a computationally efficient method for data selection, it has several limitations. First, compression-based alignment may not fully capture nuanced semantic relationships that dense representations can, potentially affecting its effectiveness for complex domains like natural language understanding, where paraphrasing is important. Second, `ZIP-FIT`'s reliance on compression means that its performance could vary depending on the nature of the textual data, particularly in highly diverse datasets where compression gains are less apparent.

# 8 DISCUSSION AND FUTURE WORK

`ZIP-FIT` introduces an efficient, embedding-free approach for data selection in language model fine-tuning. By leveraging compression algorithms to capture redundancies in data, `ZIP-FIT` enables the alignment of large-scale datasets with a target domain without the computational burden of neural embeddings. Our experiments with different compression algorithms (Figure 8) reveal that lighter compression (e.g., LZ4 at level 0) leads to better performance, achieving a 12.19% Pass@1 on HumanEval compared to 11.58% with gzip. This suggests that while compression effectively captures alignment signals, aggressive compression can remove subtle but important patterns. These insights highlight the importance of careful selection of compression parameters in optimizing the quality of data selection. Our results show that using compression-based alignment leads to faster convergence and lower cross-entropy loss compared to existing methods like DSIR and D4 (Tirumala et al., 2023; Xie et al., 2023b).

However, this approach highlights the trade-off between simplicity and the ability to capture complex semantic relationships. While compression-based methods offer a lightweight alternative, they might not fully replace embedding-based techniques for highly intricate domains, such as natural language understanding or paraphrases. Nonetheless, `ZIP-FIT`'s promising results suggest that leveraging compression as a data selection tool can be highly effective, especially in resource-constrained

scenarios and economically crucial tasks like code generation, compression can leverage the syntactic structure of the data.

Future work could explore hybrid models that combine the strengths of compression-based techniques with neural embeddings to further enhance data selection. Additionally, extending `ZIP-FIT` to support more diverse data modalities and investigating its robustness across various domains would provide a more comprehensive understanding of its capabilities and limitations. We plan for future work to study its application to complex tasks based on natural language alone and mathematics, where paraphrasing and semantics are important.

We also plan to explore the use of `ZIP-FIT` for synthetic data generation. While generating synthetic data is straightforward, selecting high-value samples for training presents challenges, especially when managing limited token budgets Villalobos et al. (2024). Autoformalization is a fantastic task for this exploration, as it inherently has a limited number of tokens, thus simulating the critical challenge of token scarcity. Additionally, studying synthetic data selection is crucial for developing self-improving agents that can avoid model collapse (Gerstgrasser et al., 2024; Kazdan et al., 2024) by ensuring high-quality data accumulation.

Furthermore, diversity was identified as an important meta-data property that can influence model performance (Miranda et al., 2024). Therefore, we aim to address this in future work by either: (1) developing an algorithm that balances diversity with alignment in data selection, or (2) creating a metric that incorporates diversity as part of its evaluation process.

[backgroundcolor=blue!10, linecolor=blue!50!black, linewidth=2pt, innertopmargin=nerbottommargin=nerrightmargin=20pt, innerleftmargin=20pt, roundcorner=10pt] **Key Takeaways:**

- **Efficiency in Data Selection:** `ZIP-FIT` utilizes compression for alignment, demonstrating significant efficiency in selecting domain-specific data, enhancing model fine-tuning.
- **Resource Optimization:** It outperforms traditional methods like DSIR, D4, and LESS by speeding up training and reducing computational demands, beneficial in resource-limited settings.
- **Domain-Specific Improvements:** Exhibits superior performance in tasks like AutoFormalization and Python code generation, where precise data alignment is crucial.
- **Practical Application:** Effective in identifying and using the most relevant data from mixed datasets, proving critical for achieving better domain-specific results.

## 9 CONCLUSION

In this work, we introduced `ZIP-FIT`, an efficient and scalable data selection method that leverages compression to enhance the downstream performance of language models for domain-specific tasks. Our experiments demonstrate that `ZIP-FIT` not only accelerates the fine-tuning process but also significantly improves downstream performance by aligning training data more closely with target tasks. By comparing against established methods like DSIR, D4 and LESS, `ZIP-FIT` proved superior in selecting highly-aligned data, especially in complex tasks such as Autoformalization and code generation. This methodology provides a resource-efficient and effective approach to data selection for model training, contributing to a better understanding of how training data impacts downstream transfer in LMs.

## IMPACT STATEMENT

`ZIP-FIT` improves task-aware data selection efficiency, reducing compute costs and enabling faster, more targeted fine-tuning. By eliminating embeddings, it enhances accessibility for low-resource settings while maintaining strong performance. Its compression-based approach offers a scalable, efficient alternative to traditional methods, with implications for sustainable AI training and fairer dataset curation.

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

## A    APPENDIX / SUPPLEMENTAL MATERIAL

Optionally include supplemental material (complete proofs, additional experiments and plots) in appendix. All such materials **SHOULD be included in the main submission.**

## B    GZIP COMPRESSION DETAILS

`gzip` is a lossless data compression algorithm that combines two primary techniques: LZ77 compression and Huffman coding. Here, we provide additional technical details on how `gzip` works.

**LZ77 Compression:**    LZ77 works by identifying repeated substrings in the input text and replacing them with backward references. Mathematically, LZ77 can be described as follows:

Given an input sequence $S = s_1, s_2, \ldots, s_n$, the algorithm searches for the longest prefix of the remaining sequence $S' = s_i, s_{i+1}, \ldots, s_n$ that matches a substring within a predefined window of previous characters. If a match is found, it is replaced by a tuple $(d, l, c)$, where:

- $d$ is the distance from the current position to the start of the matching substring,
- $l$ is the length of the matching substring, and
- $c$ is the character following the match (if any).

For example, the substring $s_i, s_{i+1}, \ldots, s_{i+l-1}$ can be replaced by the tuple $(d, l, c)$, thereby reducing redundancy in the data.

**Huffman Coding:**    After applying LZ77, `gzip` employs Huffman coding to further reduce the size of the compressed data. Huffman coding assigns variable-length codes to symbols based on their frequency of occurrence, with shorter codes assigned to more frequent symbols.

The expected length $L(X)$ of the Huffman code for a sequence of symbols $X = x_1, x_2, \ldots, x_n$ is calculated as:

$$L(X) = \sum_{i=1}^{n} p(x_i) \cdot \text{len}(C(x_i)),$$

where:

- $p(x_i)$ is the probability of symbol $x_i$,
- $\text{len}(C(x_i))$ is the length of the Huffman code for $x_i$.

This further minimizes the size of the compressed data by leveraging the statistical properties of the input.

**Combined `gzip` Compression:**    The total compressed size $C(S)$ after applying both LZ77 and Huffman coding can be approximated as the sum of the lengths of the backward references and the Huffman-coded symbols:

$$C(S) = \sum_{(d,l,c)} \text{len}(d, l, c) + \sum_{i=1}^{n} \text{len}(C(x_i)).$$

**Normalized Compression Distance (NCD):**    `gzip`'s effectiveness in data selection stems from its ability to measure the alignment between two sequences $A$ and $B$ based on how efficiently they compress together. The **Normalized Compression Distance (NCD)** is given by:

$$NCD(A, B) = \frac{C(A \oplus B) - \min(C(A), C(B))}{\max(C(A), C(B))},$$

where $C(A)$ and $C(B)$ are the compressed lengths of sequences $A$ and $B$, and $C(A \oplus B)$ is the length of the compressed concatenation of both sequences. A lower NCD indicates greater alignment between the sequences.

## B.1 Why Use Compression?

Compression algorithms, such as `gzip`, provide a computationally efficient way to detect patterns and minimize redundancy in data.

**Limitations of n-grams:** Many traditional methods, including hashed n-grams, focus on capturing immediate textual correlations by simplifying text into discrete, fixed-size buckets. Although these techniques are computationally efficient, they may not adequately capture syntactic or structural relationships within the data. Additionally, the introduce noise due to collisions during hashing.

**Challenges with Neural Embeddings:** Neural embeddings offer a powerful tool for capturing semantic relationships, but they come with significant computational costs. These embeddings are typically pre-trained on large corpora and fine-tuned for specific tasks, which requires substantial resources. Given the scalability challenges of embedding-based methods, we conjecture that a simpler method like compression can provide a more scalable and resource-efficient alternative.

We hypothesize that compression – in this case `gzip`, but perhaps a different compression algorithm –serves as a strong proxy for capturing syntactic and structural relationships in textual sequences. `gzip`'s ability to compress data based on redundancy minimization can be leveraged as a metric to align text with a target distribution.

## B.2 Composition of the Source Dataset for AutoFormalization

The source dataset for the AutoFormalization task was compiled from a variety of datasets to ensure a diverse mix of mathematical, general textual, and code-related content. Below are the details of the datasets included:

- **UDACA/AF:** 4,300 samples from informal formalization statements.
- **C4:** 10,000 samples from the clean crawl of the internet, ensuring a broad linguistic variety.
- **LeanDojo:** 10,000 samples from task-oriented proofs and tactics.
- **LeanDojo Informalized:** 10,000 samples combining traced tactics with informal descriptions, aiming to bridge formal reasoning and natural language.
- **UDACA/AF-split:** 10,000 samples, a variant of the UDACA/AF dataset with split annotations.
- **WikiText:** 10,000 samples from a collection of professionally curated articles, providing a rich linguistic framework.
- **Algebraic Stack:** Samples from various subsets of mathematical and programming languages, capped at 10,000 samples per subset or fewer if the total subset size was under this threshold.

Each dataset was selected to complement the others by covering different aspects of language use, from technical to informal, ensuring the model's exposure to a wide range of linguistic structures and contents. The total dataset size aggregated to approximately 185,000 sequences, which were then subjected to alignment scoring and further processing for model training.

## B.3 Composition of the Source Dataset for Code Generation

The source dataset for the Code Generation task was assembled from various data sources to provide a diverse range of coding and natural language contexts. Below are the details of the datasets included:

- **MBPP (Google Research):** A total of 964 samples focusing on Python coding challenges.
- **Python Code Instructions (18k Alpaca):** 5,000 sequences providing natural language prompts for Python code, fostering a practical approach to code generation.
- **Python Docstrings (Calum/The Stack):** 5,000 sequences each of Python function docstrings integrating detailed natural language documentation of python functions.

- **Python Docstrings (Calum/The Stack):** 5,000 sequences each of Python function code bodies, integrating raw python code without documentation.
- **C4 (AllenAI):** 10,000 samples from a clean web crawl.
- **WikiText:** 10,000 samples from a collection of curated articles, providing rich natural language training material.
- **Algebraic Stack:** A selection of sequences from various programming language subsets, each capped at 10,000 samples or the total subset size if less than this threshold.

This combination of datasets was specifically chosen to challenge our methods 's ability to choose syntactically correct and functionally accurate Python code, while also responding appropriately to natural language prompts.

### B.4 Hyperparameters for Model Fine-Tuning

All models in our experiments were fine-tuned with the following unified setup, aimed at ensuring a consistent evaluation across different models and data selection strategies.

**Models and Tokenizer:** The fine-tuning was performed using the following models:

- InterLM-Math-Plus-1.8B
- Gemma-2-2B
- Mistral7B

**Training Settings:** The key hyperparameters used across all models are as follows:

- **Block Size:** 1024 tokens
- **Learning Rate:** $7.5 \times 10^{-7}$
- **Batch Size:** 4 (per device)
- **Number of Epochs:** 1
- **Weight Decay:** 0.01
- **Maximum Gradient Norm:** 1.0

Training was facilitated using the `Trainer` class from Hugging Face's Transformers library, with the Accelerate library handling model parallelism to efficiently utilize available computational resources.

**Evaluation Metrics:** For model evaluation, we employed:

- **Cross-Entropy Loss** at the end of training to measure the effectiveness of the fine-tuning.

Fine-tuning was performed under controlled conditions to ensure fair comparison between data selected by `ZIP-FIT`, DSIR, and manual curation methods. The effectiveness of each method was assessed based on how the models performed on the ProofNet and HumanEval.

**Data Handling and Logging:** All logs, model checkpoints, and tokenizer settings were systematically saved in designated directories for thorough analysis post-experiment

This comprehensive and standardized approach to fine-tuning ensures that our experimental results are robust, reproducible, and transparent, providing clear insights into the effectiveness of the data selection methodologies employed in our study.

## C Rationale for the Method Name ZIP-FIT

We chose the name **ZIP-FIT** for two reasons:

1. **ZIP** refers to the use of `gzip` compression for data selection, where compression aligns the data for better future fine-tuning (or **FITting**).

2. The name also references scaling laws, as `ZIP-FIT` consistently reduces loss faster than competing methods, implying better power-law scaling parameters, drawing a parallel to **Zipf's law** Piantadosi (2014), which describes similar scaling behavior in language models.

*Remark:* Zipf's law Piantadosi (2014) describes the inverse relationship (thus power law $f(r) \propto 1/r^s$, where $r$ is the rank and $f(r)$ is the frequency of the word with rank $r$) between a word's frequency and its rank in natural language, a pattern that reflects scaling behavior. Rank in this context is the position of the word after sorting with respect to frequency in the text.

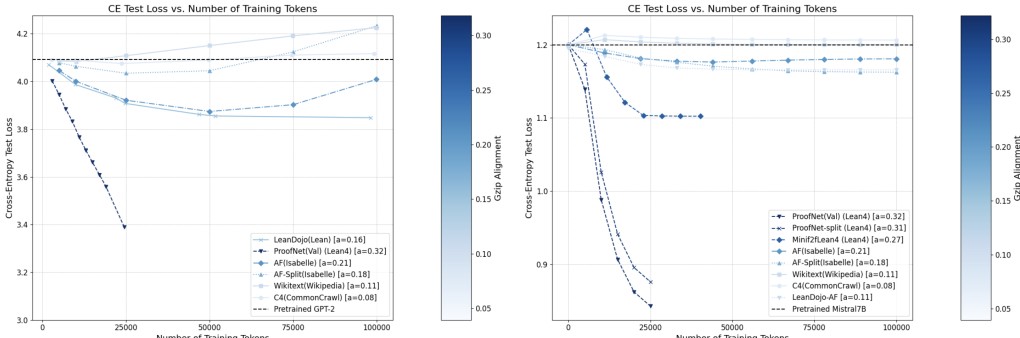

Figure 6: **Highly aligned data lowers cross-entropy loss more efficiently.** The x-axis shows the number of training tokens, and the y-axis represents the cross-entropy (CE) test loss on the ProofNet test set. Different curves correspond to datasets filtered by different alignment scores, indicating their relevance to the target domain. The most aligned data reduce Test CE loss significantly faster than less aligned data. The left panel depicts results using GPT-2, and the right panel uses Mistral7B, demonstrating that using highly aligned data not only accelerates training but also achieves better model performance, validating the effectiveness of ZIP-FIT for data selection in fine-tuning.

## D  HIGHER ALIGNMENT LEADS TO MORE EFFICIENT TRAINING

**Experiment:** We fine-tuned GPT-2 (124M) and Mistral7B for the AutoFormalization task using different datasets scored with ZIP-FIT alignment. We used ProofNet (test) for the evaluation. The curves represent different datasets with varying alignment to the target domain (ProofNet validation).

**Results:** More aligned data reduces CE loss quickest, as shown by the steep decline for high-alignment datasets. This is most evident as ProofNet (validation). Less aligned data require significantly more tokens to achieve similar performance. This demonstrates that targeted data selection with ZIP-FIT accelerates fine-tuning and improves performance, reducing computational costs.

# E   ADDITIONAL EXPERIMENTAL RESULTS: DATA SELECTION FOR EFFICIENT FINE-TUNING USING ZIP-FIT

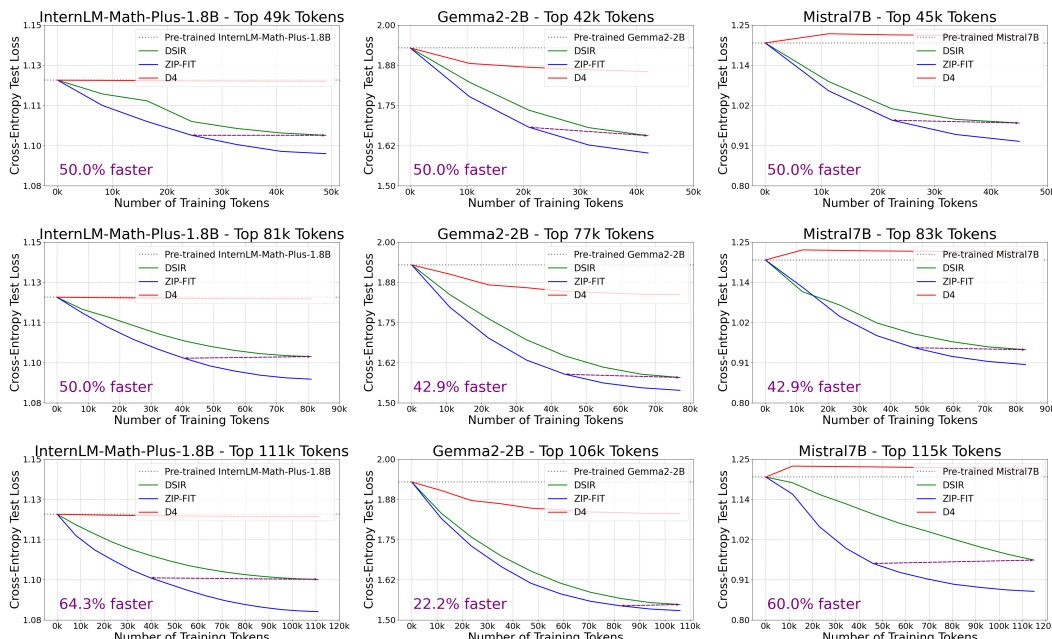

Figure 7: **ZIP-FIT consistently achieves a lower test loss at a faster rate compared to D4 and DSIR for Autoformalization**. The plots show the cross-entropy test loss against the number of training tokens for three models (InterLM-Math-Plus-1.8B, Gemma-2-2B, and Mistral7B) across various token selection sizes. ZIP-FIT (blue line) consistently surpasses both DSIR (green line) and D4 (red line) across all model and token size configurations, emphasizing its superior data processing efficiency. The percentage labels in each plot denote the relative speedup of ZIP-FIT over DSIR in attaining the lowest cross-entropy loss, further underscoring the method's scalability and adaptability for domain-specific fine-tuning.

# F BASELINE COMPARISON USING TEACHER-FORCED ACCURACY(TFA) FOR AUTOFORMALIZATION

**Teacher-Forced Accuracy (TFA)**: TFA evaluates mathematical reasoning tasks by measuring exact syntactic alignment with reference solutions (Jiang et al., 2023a). It correlates strongly with correctness in competition-level mathematics (Gulati et al., 2024), making it a suitable metric for autoformalization.

We evaluated Teacher-Forced Accuracy (TFA) on ProofNet's test split by fine-tuning Gemma-2-2B on datasets curated with ZIP-FIT, DSIR, and LESS.

Table 3: **Performance and efficiency comparison of data selection methods.** Results show Pass@1 and Pass@10 scores on HumanEval using top 1M tokens for fine-tuning, along with data selection time. Data selection times exclude fine-tuning time.

| Fine-tuning | Data Selection | Pass@1 (%) | Pass@10 (%) | Selection Time |
|---|---|---|---|---|
| None | Pre-trained Gemma-2-2B | 15.24 | 38.81 | – |
| None | Pre-trained Gemma-2-2B (4-bit quantized) | 6.09 | – | – |
| Full FT | ZIP-FIT | **18.86** | 41.78 | **32s** |
| Full FT | LESS | 18.06 | 40.19 | 19h |
| Full FT | DSIR | 17.98 | **44.27** | 97s |
| QLoRA | ZIP-FIT | **12.19** | – | **32s** |
| QLoRA | DSIR | 9.14 | – | 97s |
| QLoRA | D4 | 6.09 | – | 7h 40m |

## G  IMPACT OF COMPRESSION ALGORITHMS AND LEVELS

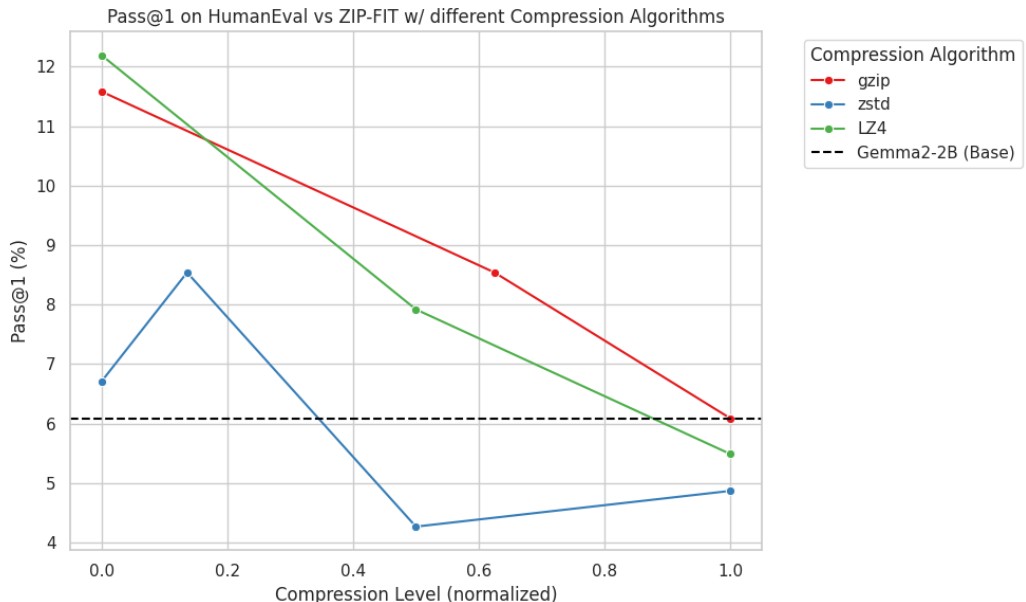

Figure 8: **Lighter compression preserves crucial information for data selection.** At minimum compression levels, both gzip and LZ4 achieve the strongest Pass@1 scores (11.58% and 12.19%), significantly outperforming the base model (6.09%, dashed line). Performance systematically degrades with increased compression across all algorithms, suggesting that aggressive compression removes valuable alignment signals.

To investigate the impact of different compression algorithms on ZIP-FIT's performance, we conducted experiments comparing three widely used compression methods: gzip, zstd, and LZ4. Each algorithm was tested across its available compression levels, normalized to a 0-1 scale for comparison. As shown in Figure 8, compression algorithm choice and level significantly impact performance.

Key findings include:

- LZ4 at minimum compression achieves the best performance (12.19% Pass@1)
- Higher compression levels generally lead to decreased performance across all algorithms
- gzip shows more stable performance degradation compared to LZ4 and zstd
- zstd consistently underperforms relative to both GZIP and LZ4

These results suggest that lighter compression better preserves the structural information needed for effective data selection. The superior performance of LZ4 at minimal compression indicates that aggressive data compression may remove subtle but important patterns useful for alignment assessment.

# H    DATA SELECTION PROFILING (RUN TIMES)

`ZIP-FIT` performs selection up to 65.8% faster than DSIR and 21,076% (=5h/85s=211, which is 2 orders of magnitude) faster than D4. Experimental results comparing `ZIP-FIT` vs DSIR profiling/run time for Code data selection can be found in figure 9. Note that depending on the dataset and number of samples these numbers may not hold. Compression may not scale well to long-context datasets and depending on the source dataset, our run times varied widely. However, on average we observed that `ZIP-FIT` is comparable to DSIR and generally faster. More experiments across a wider range of datasets need to be conducted in order to infer more.

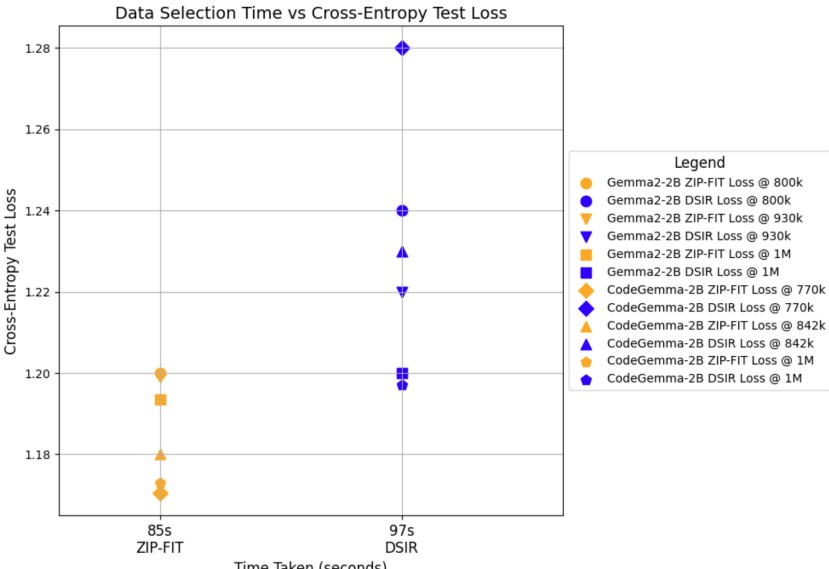

Figure 9: **`ZIP-FIP` demonstrates lower cross-entropy and lower run time during data selection than competing DSIR and D4 methods.** `ZIP-FIT` is cheaper, faster, and better performing. The run times do no include fine-tuning time, since it's a constant offset across all models. D4's data selection (not shown) takes 5hs because it uses an embedding model (opt-125m Zhang et al. (2022)), the same one as the original paper Tirumala et al. (2023).

# I  NORMALIZED COMPRESSION DISTANCE (NCD): HOW COMPRESSION MEASURES DATA ALIGNMENT INTUITIVELY

Normalized Compression Distance (NCD) uses compression to quantify how much two objects $x$ and $x'$ share in common. First, observe that if $x$ and $x'$ are similar, then the compressed size of their concatenation, $C(x \oplus x')$, will not be much larger than the compressed size of one of them alone—because most of their information overlaps. Conversely, if $x$ and $x'$ are very different, $C(x \oplus x')$ will be roughly the sum of their individual compressed sizes, since they share little or no redundancy.

Formally, NCD isolates how much extra information one object contributes beyond their shared core by subtracting $\min(C(x), C(x'))$ from $C(x \oplus x')$. This "delta" measures what is new or unique in combining $x$ with $x'$. Dividing that delta by $\max(C(x), C(x'))$ then normalizes the distance so that its value lies between 0 and 1:

$$\text{NCD}(x, x') \;=\; \frac{C(x \oplus x') - \min\big(C(x),\, C(x')\big)}{\max\big(C(x),\, C(x')\big)}.$$

When $x$ and $x'$ are identical, the numerator is near zero, giving NCD $\approx 0$. When $x$ and $x'$ are very different, the numerator is comparable to $\max(C(x), C(x'))$, so NCD $\approx 1$. By measuring how easily one object's information can be "compressed away" given the other object, NCD succinctly captures their similarity or dissimilarity.

# J  QUALITATIVE ANALYSIS

Qualitative results show top 20 examples can be found it table **??**.

**Selected Samples by `ZIP-FIT` with `ZIP-FIT` Alignment scores**

| Sample Text (Beginning) | Alignment Score |
|---|---|
| Across all his bands and projects, Townsend has released twenty @-@ three studio albums and three live albums. | 0.5000 |
| Require Import CodeDeps. Require Import Ident. Local Open Scope Z_scope. Definition _addr := 1%positive. Definition _g := 2%positive. | 0.4928 |
| This Photostock Vector Night Sky Background With Full Moon Clouds And Stars Vector Ilgraphicration has 1560 x 1560 pixel resolution... | 0.4926 |
| module Structure.Logic where ... | 0.4926 |
| { dg-do compile } PR fortran/51993 Code contributed by Sebastien Bardeau `<bardeau at iram dot fr>` module mymod type :: mytyp... | 0.4891 |
| For over ten years, the St. Louis Mercy home has formed a special connection with a local community theatre: The Muny. This summer the... | 0.4889 |
| Read("SchreierSims.gi"); LoadPackage("AtlasRep"); MicroSeconds := function() local t; t := IO_gettimeofday(); return t.tv_sec * 1000000 + t.t | 0.4889 |
| Get the keyId used by this peer (this peer's identifier). This is stored in the key store. | 0.4857 |
| Initializes and adds a node to the graph. NOTE: At least the type must be supplied for the Node to exist in the graph. Args: graph: The graph... | 0.4853 |
| def bgra2rgb(img): cv2.cvtColor(img, cv2.COLOR_BGRA2BGR) has an issue removing the alpha channel, this gets rid of wrong trans... | 0.4853 |

Table 4: Beginning characters of the top 20 samples selected by ZIP-FIT when the target task is code generation.

**Selected Samples by DSIR with `ZIP-FIT` Alignment scores**

| Sample Text (Beginning) | `ZIP-FIT` Alignment Score |
|---|---|
| <a href="https://colab.research.google.com/github/julianovale/simulaca o_python/blob/master/0006_ex_trem_kronecker_algebra_computacao ... | 0.122 |
| library(qcc) \\n death=c(2,1,2,4,2,5,3,3,5,6,3,8,3,3,6,3,6,5,3,5,2,6,2,3,4, 3,2,9,2,2,3,2,10,7,9,6,2,1,2,4,2,5,3,3,5,6,3,8,3,3,6,3,6,5,3,5,2,6,2 ... | 0.121 |
| gap >List(SymmetricGroup(4), p - >Permuted([1 .. 4], p)); \\n perms(4); [ [ 1, 2, 3, 4 ], [ 4, 2, 3, 1 ], [ 2, 4, 3, 1 ], [ 3, 2, 4, 1 ... | 0.191 |
| # Solutions \\n ## Question 1 \\n >'1'. Using a 'for' loop print the types of the variables in each of the >following iterables: \\n >'1' ... | 0.145 |
| # Some small pregroups \\n # The lists of small pregroups were generated by \\n # Chris Jefferson <caj21@st-andrews.ac.uk> and \\n ... | 0.195 |
| adjacency_mat = [ false true true true true true true true true false true true true true false false false true false true false true false ... | 0.182 |
| \section{Lookup table used for accessing child voxels using a parent's child descriptor} \label{app:lookup-table} \lstset{language=C,cap ... | 0.199 |
| \* statistics test_nist.c \\n * \\n * Copyright (C) 1996, 1997, 1998, 1999, 2000, 2007 Jim Davies, Brian Gough \\n * \\n * This pro ... | 0.180 |
| Problem Description Write a python function to find the first missing positive number. \\n def first_Missing_Positive(arr,n): \\n ptr = 0 ... | 0.239 |
| import numpy as np \\n mandelTable = [[0,0,0,0,0,0,0,0,0,0,0,0,0,0,0,0,0,0,0,0,0,0,0,0,0,0,0,0,0,0,0,0,0,0,0, ... | 0.189 |

Table 5: Beginning characters of the top 20 samples selected by DSIR when the target task is code generation. DSIR does not easily provide alignment scores, so instead we report the `ZIP-FIT` scores, which reveals that `ZIP-FIT` doesn't score highly the DSIR examples which might explain why `ZIP-FIT` achieves better CE loss.

## J.1 CHOICE OF EXAMPLES

To evaluate our approach, we selected **three examples from MiniF2F in Lean4**. ProofNet was avoided since its validation and test sets are highly similar, meaning the model could memorize and correctly predict the first few examples without true generalization. MiniF2F, containing formalized math exercises from textbooks, provided a more robust test set. Some exercises differ only slightly (e.g., proving a theorem for $+$ instead of $*$), making it a better challenge for evaluating autoformalization models.

## K PASS@K FOR AUTOFORMALIZATION

The following prompt was used for evaluating syntax error compilation Pass@k in autoformalization experiments:

```
from textwrap import dedent

def my_prompt_format(nl_stmt: str) -> str:
    """Format a prompt to translate a natural-language math statement to Lean."""
    prompt: str = dedent(f"""\
        Your task is to translate the natural-language mathematical statement
        into a formal Lean statement using the following format:

        natural language statement:
        Let $z=\\frac{{1+i}}{{\\sqrt{{2}}}}.$ What is ...

        formal Lean language statement:
        ##
        theorem amc12a_2019_p21 (z : ) (h : z = (1 + Complex.I) / Real.sqrt 2) :
```

```
        ( k in Finset.Icc 1 12, (z^(k^2))) * ( k in Finset.Icc 1 12, (1 / z^(k^2))
        sorry
    ##
    ...

    natural language statement:
    {nl_stmt}

    formal Lean language statement:
    """)
return prompt
```

## L  TEACHER FORCED ACCURACY (TFA) RESULTS

| Method | TFA (%) | Time | Hardware |
|--------|---------|------|----------|
| Gemma-2-2B | 46.73 | – | – |
| ZIP-FIT | $61.38 \pm 2.73$ | **79s** | CPU |
| DSIR | $59.29 \pm 1.33$ | 135s | CPU |
| LESS | $63.72 \pm 3.75$ | 20h 45m | 4 A100-80GB |

Table 6: AutoFormalization: `ZIP-FIT` achieves similar Teacher Forced Accuracy (TFA) compared to LESS in a statistically significant way. Confidence interval are at 95%.

For Gemma-2-2B, Table **??** shows that `ZIP-FIT` achieves a TFA of 61.38% ± 2.73%, which is comparable to LESS (63.72% ± 3.75%) but with a drastically reduced selection time of 79 seconds compared to LESS's 20 hours and 45 minutes. `ZIP-FIT` also outperforms DSIR in both TFA (59.29% ± 1.33%) and selection speed (135 seconds).

## M  FUTURE WORK (CONT.)

**Lossless Compression for Alignment:** While `ZIP-FIT` has demonstrated substantial efficiency for data selection, there are several promising directions for future exploration. One potential enhancement is leveraging faster compression algorithms, such as `LZ4` and `Snappy`, which offer rapid processing speeds at the cost of lossy compression. In our current approach, we utilize `gzip` for compression-based alignment, which is lossless and provides a robust foundation. However, `LZ4` and `Snappy` are optimized for speed and could potentially offer even greater computational efficiency without the need for decompression in our pipeline. Given that our primary goal is efficient data selection rather than perfect data recovery, these faster algorithms might be more suitable.

**Autonomous Validation Set Generation**: A current limitation of ZIP-FIT is its dependence on a small, curated validation set (e.g., 185 samples for ProofNet and 82 samples for half the HumanEval test set). Future work could explore the use of generative models to create synthetic validation sets from task-specific instructions. This approach could also be expanded to enable autonomous self-directed, model-driven generation of validation data.

