# OpenReview forum: "ZIP-FIT: Embedding-Free Data Selection via Compression-Based Alignment for Code"
_ICLR.cc/2026/Conference — Submitted to ICLR 2026_

### Official Review · Reviewer_DPeZ · 2025-10-28

**Soundness:** 1
**Presentation:** 2
**Contribution:** 2
**Rating:** 2
**Confidence:** 4

**Summary:**

This paper proposes an efficient and embedding-free data selection method inspired by gzip compression. By comparing with related methods on code generation and autoformalization, the authors demonstrate that the proposed ZIP-FIT is simple and effective.

**Strengths:**

1. The proposed method, inspired by gzip compression, is simple yet effective.
2. ZIP-FIT demonstrates better performance and faster convergence speed compared with DSIR, D4, LESS.

**Weaknesses:**

1. This paper only evaluates ZIP-FIT on program synthesis tasks. The proposed ZIP-FIT is a task-agnostic method, yet the authors only evaluate it on one task, which significantly limits the generalization of ZIP-FIT.
2. Even for these two program synthesis tasks, the evaluation is not comprehensive and clear. For the limitation of comprehensiveness, the authors only evaluate on two datasets, HumanEval and ProofNet. Besides, the authors only show the results of one LLM, Gemma-2b. For example, the authors only show the pass@1 rates of Gemma-2-2b on HumanEval, without CodeGemma-2b. For the limitation of clear experiment settings, the authors do not reveal the details about training and prompting. For example, what is the setting in the first row in Table 1 and Table 2? Zero-shot performance? Or fine-tuning with full datasets?
3. As the authors admit that ZIP-FIT is negatively influenced by the diversity of original datasets (I don't take this point as a weakness of the paper), there is no analysis on how much it is influenced by the diversity. A quantized relationship is required for estimation in real-world applications.

**Questions:**

There are some missing references for tables and LaTeX compilation errors in the paper. Please have a check.

---

### Official Review · Reviewer_Z1tU · 2025-10-30

**Soundness:** 2
**Presentation:** 3
**Contribution:** 3
**Rating:** 4
**Confidence:** 4

**Summary:**

The paper introduces a simple but effective method to select an aligned set of data from a large corpus to align language models to specific tasks. Instead of embeddings, n-grams or losses, the normalized compression distance is used. This allows fast data selection on CPU, and the experiments show that highly aligned data is selected on structured domains.

**Strengths:**

The method is simple, fast, and can be confidently applied to any domain with strong syntactic structures. Experiments show strong performance on domains where the task involves heavily structured elements (like code).

With the exception of some missing table references (line 255, line 290) and citations style (I'd cite a (method, author) combination as `method (authors)` instead of `method authors`, for example, `Lean4 (Moura et al. (2015))` on line 163), the presentation is clear and easy to follow.

**Weaknesses:**

The paper considers "only" two benchmarks with highly specific domains. Given that the number of domains is small, it is reasonable to implement an even more domain-specific approach, for example, a simple regex test for finding `def \w+\(` can find documents with Python code, on CPU, without requiring compression. Complex domain combinations (like Multipl-E [1]) will allow to make a more definitive claim of the relevance of this method.

[1] Cassano, F., Gouwar, J., Nguyen, D., Nguyen, S., Phipps-Costin, L., Pinckney, D., ... & Jangda, A. (2023). Multipl-e: A scalable and polyglot approach to benchmarking neural code generation. IEEE Transactions on Software Engineering, 49(7), 3675-3691.

**Questions:**

Did you consider any experiments on plain-text tasks? Disregarding subtle semantics, I can imagine natural language structures the compress well, and ZIP-FIT might work better than expected.

Minor:
* There's some \latex stuff going on on line 452.
* As mentioned, there's some missing table references.

---

### Official Review · Reviewer_W6Xv · 2025-10-31

**Soundness:** 3
**Presentation:** 2
**Contribution:** 3
**Rating:** 6
**Confidence:** 3

**Summary:**

The paper proposes ZIP-Fit, a compression-based data-selection method for domain-specific fine-tuning that avoids the use of embeddings or gradient features. It measures alignment between source and target data via Normalized Compression Distance, capturing syntactic and structural similarity. This enables efficient, task-relevant data selection, yielding better downstream accuracy, faster convergence, and lower compute cost than DSIR, D4, or LESS. Experiments on Python code generation and Lean4 autoformalization show higher Pass@1 and Pass@5 scores and much faster data selection and training.

**Strengths:**

+ The investigated problem is timely, and the proposed ZIP-FIT method is refreshingly simple: compute a compression-based alignment score between each candidate source sample and the target examples, rank, and take top-K. There’s no embedding model to train, no gradient logging, no GPU-heavy similarity search.
+ The paper demonstrates that smaller, well-aligned subsets selected by ZIP-FIT train faster and land at lower cross-entropy than larger but noisier mixtures.
+ Experiments demonstrate strong downstream metrics on the HumanEval and math autoformalization.
+ Detailed analysis. The paper doesn’t just report numbers. It correlates ZIP-FIT alignment scores with downstream cross-entropy loss across datasets and shows that higher compression-based alignment predicts lower loss after fine-tuning.

**Weaknesses:**

- The scope of domains might be limited.  The paper itself acknowledges that compression-based alignment may miss higher-level paraphrastic/semantic similarity in natural language, where style can vary without changing meaning.
- There are some presentation issues, including:
- [line 091] “code-geneation” -> generation
- [Fig. 4, Table 2] “AutoFormalization” or “Autoformalization”?
- [line 255] “Table ??”
- mixed usage of ‘k/K’, e.g., “353K”, “353k”, “695k”, etc.

**Questions:**

- Q1: How exactly are target splits chosen for alignment?

---

### Official Review · Reviewer_JhTj · 2025-11-01

**Soundness:** 3
**Presentation:** 2
**Contribution:** 3
**Rating:** 6
**Confidence:** 3

**Summary:**

ZIP-FIT is a simple, embedding-free data selection method for LM fine-tuning that uses compression to measure how well each candidate training example aligns with a small set of target examples: compute a Normalized Compression Distance (NCD)–based alignment, rank the source pool, and train on the top-K subset. The authors show this one-shot, CPU-only selector yields faster convergence and better downstream results than DSIR/D4 (and other baselines) on code generation and autoformalization—e.g., 18.86% Pass@1 on HumanEval in 32 s and 14.0% Pass@5 on Lean4, while running up to 65.8% faster than DSIR. The intuition is that structurally similar texts compress well together (low NCD), so compression acts as a cheap proxy for task alignment

**Strengths:**

ZIP-fit presents a resource efficient data selection technique that is embedding free and faster as compared to prior works. Further the proposed technique outperforms baselines in Lean4 and HumanEval Python generation over various open-source models.

**Weaknesses:**

- The authors mention embedding-based (like BERT), and Continual pre-training-based selection metrics, but do not compare against them. Hence, the benefits of the technique cannot be truly seen against embedding-based baselines other than just that the proposed technique is computationally efficient.
-  The paper does not show a finetune on all as a reference curve that the ZIP-fit method can be compared against
- The zip-fit approach depends on the curation of a high-quality representative benchmark set. Further, there can be multiple ways towards solving the same problem (for instance, using for loops against while loops in Python and writing lambda functions). Hence, ZIP-fit selected data might not contribute towards learning multiple approaches of solving the same problem.


Missing citations: In various places citations are missing. Line 255, 290, 1215, 1320
Formatting: The appendix is incorrectly formatted, and the discussion section contains some formatting code that the authors forgot to remove.

**Questions:**

Please look at the weaknesses.

Other questions:
- What is the performance of a naive uniform/random sampling baseline on both tasks ?
- What is the comparison with cross-entropy–difference (Moore–Lewis) selection ?

---

### Official Review · Reviewer_CJZu · 2025-11-07

**Soundness:** 3
**Presentation:** 2
**Contribution:** 2
**Rating:** 4
**Confidence:** 3

**Summary:**

The paper introduces ZIP-FIT, a novel embedding-free data selection framework for fine-tuning large language models (LLMs), particularly in domain-specific tasks such as autoformalization and code generation.

**Strengths:**

The paper’s core insight — using compression as a proxy for task alignment — is conceptually elegant and computationally lightweight. Extensive experiments on two challenging domains (Autoformalization and Python Code Generation) demonstrate consistent improvements in both convergence speed and downstream performance. And the paper is well-written and easy to follow.

**Weaknesses:**

1. While the method performs impressively on code and mathematical text, it remains unclear how well it generalizes to open-domain natural language tasks, where semantics and paraphrasing are more complex and less syntactically driven.
2. ProofNet has limited linguistic variation, and its syntax regularity may inflate alignment correlations (R² = 0.9).
3. Limited Benchmark Scope: the two domain Autoformalization and Python code generation in the experiments, share a highly structured syntax and low semantic ambiguity, making them especially favorable to compression-based similarity metrics..

**Questions:**

Please refer to the weakness.

---

### Meta-Review · Area_Chair_imNz · 2026-01-06

**Summary:**

This paper proposes ZIP-FIT, a data selection framework that uses compression for data selection using any embedding models. The paper defines the Normalized Compression Distance (NCD) that is computed between two inputs based on the length of the compressed size of the concatenation of the two inputs. The distance is small if the two inputs are very similar and hence their concatenation can be compressed to a similar size. NCD is used to select the top-k examples from the dataset based on the highest alignment scores.

The reviewers highlight strengths such as the simplicity and effectiveness of the proposed method.

**Reviewer Concerns:**

- **Limited experiments (raised by all reviewers)**: The proposed ZIP-FIT is a task-agnostic method, yet the authors only evaluate it on one task, which significantly limits the generalization of ZIP-FIT. The evaluation is also limited to only two datasets HumanEval and ProofNet.
- **Formatting errors (raised by all reviewers)**: There are some missing references for tables and LaTeX compilation errors in the paper.
- **Missing details (DPeZ)**: The details of experimental settings are not clear.
- **Missing analysis on influence by diversity (DPeZ)**.
- **Missing experiments (JhTj)**: The authors mention embedding-based (like BERT), and Continual pre-training-based selection metrics, but do not compare against them. The paper does not show a finetune on all as a reference curve that the ZIP-fit method can be compared against.

**Reviewer Scores:**

The reviewers gave scores 2, 4, 4, 6, 6. The authors did not respond during the rebuttal and did not revise the paper. The reviewers share concerns regarding limited experiments, missing details, and formatting issues. The AC acknowledges that the idea of data selection using compression methods is interesting and has potential, and agrees with reviewers that important concerns need to be resolved to demonstrate the generality of the method and the presentation.

---

### Decision · Program_Chairs · 2026-01-26

Reject